# Magnitude of tuberculosis treatment outcomes and associated factors in public health institutions of Arba Minch town, Southern Ethiopia: A multi-centered retrospective cross-sectional study

Behailu Asmamaw[1], Temesgen Tamiru Tadese[1], Rediet Yifru Mamo[1], Berhanu Amare Taye[2], Abrham Workineh Azale[2], Yohannes Chemere Wondmeneh[3], Dagmawi Nega Shibeshi[4], Eunice Borkor Bortequaye[5], Temesgen Teklu Ziza[2,6], Temesgen Esayas Bolliso[7], Tiwabwork Tekalign[8], Awoke Guadie[9]*

1 School of Medicine, College of Medicine and Health Sciences, Arba Minch University, Arba Minch, Ethiopia, 2 School of Medicine, College of Medicine and Health Sciences, Addis Ababa University, Addis Ababa, Ethiopia, 3 School of Medicine, College of Medicine and Health Sciences, University of Gondar, Gondar, Ethiopia, 4 Department of Internal Medicine, Yekatit 12 Medical College, Addis Ababa, Ethiopia, 5 School of Medicine, University for Developmental Studies, Tamale, Ghana, 6 Rollins School of Public Health, Emory University, Atlanta, Georgia, United States of America, 7 School of Medicine, College of Medicine and Health Science, Hawassa University, Awasa, Ethiopia, 8 School of Nursing, College of Medicine and Health Sciences, Arba Minch University, Arba Minch, Ethiopia, 9 Department of Biology, College of Natural Sciences, Arba Minch University, Arba Minch, Ethiopia

* awokeg@yahoo.com

## Abstract

### Background

In Ethiopia, the incidence rate of tuberculosis (TB) has been steadily increasing, from 119 per 100,000 in 2021–126 per 100,000 in 2022 and 146 per 100,000 in 2023. Moreover, TB remains the second leading cause of death after malaria, and the third leading cause of hospital admissions. So, the aim of this study is to determine the treatment outcome and associated factors in the public health institutions of Arba Minch town.

### Methods

A multi-centered retrospective cross-sectional study was conducted involving 609 tuberculosis patients admitted from September 2021-August 2024 at public health institutions in Arba Minch town. A structured data extraction form was used by trained research assistants to collect the data from TB patient registration record books. Variables with a p-value <0.25 in binary logistic regression were further analyzed using multivariable logistic regression. Statistically significant factors were considered those with a p-value <0.05.

**Data availability statement:** All relevant data are within the manuscript.

**Funding:** The author(s) received no specific funding for this work.

**Competing interests:** The authors have declared that no competing interests exist.

## Result

The majority (53.7%) of the participants were aged between 21 and 40 years. Most participants were diagnosed with pulmonary tuberculosis (71.8%) and found to be new (92.9%) patients. According to this study, the magnitude of successful treatment outcomes was found to be 86.9%. In the multivariate logistic regression, being unmarried (p = 0.023), educational level (p = 0.028), and having extra-pulmonary TB (p = 0.017) have been found significantly associated with successful treatment outcome.

## Conclusion

The study indicates a relatively positive rate of successful treatment outcomes for TB. Although the treatment outcome results are positive, targeted interventions are needed for individuals who are married, have a low educational status, and have been diagnosed with pulmonary tuberculosis.

## Introduction

Tuberculosis (TB) has continued to be one of the major public concerns worldwide, being the most fatal infectious disease, killing nearly 1.25 million people in 2023. According to the 2024 WHO TB report, it is estimated that nearly 10.8 million people are affected with TB, among these people living in the 30 nations labeled as high-burden countries account for nearly 90% of the cases [1]. It is a disease that disproportionately affects people living in resource-limited settings [2]. Low- and middle-income countries, including Sub-Saharan Africa, account for 94% of TB-related illnesses and deaths [3]. Tuberculosis is both a preventable and curable disease that is caused by the intracellular pathogen, *Mycobacterium tuberculosis.* Although TB mainly affects the lungs (pulmonary TB), it can also involve any other organ in the body like lymph nodes, pleura, intestine, bone and soft tissues, and central nervous system which are collectively termed as extra-pulmonary TB (EPTB) [4].

Ethiopia, a country in Eastern Africa, is among the 30 high-burden countries for TB and TB/HIV, with an estimated incidence rate of 146 per 100,000 population (188,000 cases) and 25,000 TB-related deaths in HIV-negative individuals (19 per 100,000). The incidence rate has been steadily increasing from 119 per 100,000 in 2021–126 per 100,000 in 2022 and 146 per 100,000 in 2023 [5]. Moreover, TB remains the second leading cause of death after malaria, and the third leading cause of hospital admissions following deliveries and malaria, according to the Ethiopian Federal Ministry of Health (FMOH) [6]. These trends highlight TB as a serious public health concern that requires urgent intervention.

The treatment success rate (TSR) for TB varies across African nations, ranging from 53% in Nigeria to 92% in Ethiopia (2010–2020) [7]. According to the Global TB Report 2024, the Ethiopian (national) TSR among newly diagnosed TB patients is 86%, and 77% for patients with HIV co-infection, which is lower than the WHO African and global averages and the target set by the end TB strategy [1]. The TSR also

varies significantly among different regions of Ethiopia, ranging from 82.5% in western Ethiopia to 92% in Eastern Ethiopia [8–12]. Previous studies showed that being male, having HIV co-infection, age > 45 years, rural residence, underweight, retreatment, smear-negative cases, patients with no adherence support, limited access to transport, distance from home to a treatment center, and low income were factors associated with poor treatment outcomes [8–13].

According to the end TB strategy, set a milestone for the year 2025, reaching the target of 90% TSR needs intensive research efforts and rigorous monitoring and evaluation of outcomes [14]. While Ethiopia has made significant progress in TB control, there is limited evidence on region-specific treatment outcomes and associated factors. Regular monitoring of outcomes and predicting factors is of paramount significance regarding addressing specific challenges faced by different population segments and making tailored decisions. There is limited data on TB treatment outcomes and associated factors in Southern Ethiopia, specifically in the Gamo zone, Arba Minch town. This study aims to fill this gap by identifying key determinants of treatment outcomes, which can inform targeted interventions to improve TB care and reduce disease burden.

Therefore, the objective of this study is to assess the treatment outcomes and associated factors among TB patients in public health institutions of Arba Minch town, Southern Ethiopia.

## Methods

### Description of the study area

This facility-based retrospective study included TB patients treated between September 2021 and August 2024 to assess the magnitude of treatment outcomes and associated factors among patients with confirmed TB in Arba Minch town, Southern Ethiopia. Gamo zone is one of the six zones and five special woredas (districts) which are found in the Southern Ethiopia Regional State, Arba Minch being its major town. Arba Minch town is located 505 km south of Addis Ababa, the capital city of Ethiopia, and has an estimated population size of more than 200,000 people (with an estimated population of male 101,000 and female 100,000) [15]. In Arba Minch town, there are two hospitals (Dilfana Primary Hospital and Arba Minch General Hospital) and one health center (Secha Health Center).

### Study population and criteria for inclusion and exclusion

All TB patients who were diagnosed with and treated for TB at Dilfana Primary Hospital, Arba Minch General Hospital, and Secha Health Center were the source population.

### Eligibility criteria

**Inclusion criteria.** The data of all patients diagnosed with TB and who were initiated on anti-TB treatment in public health institutions of Arba Minch town during the study period were included in the study. Patients, who had full information in the unit TB register including clearly documented treatment outcomes, were considered eligible.

**Exclusion criteria.** Patients who were referred to other health facilities and/or had incomplete information were excluded from the study. Records were considered incomplete if they missed data on essential variables such as age, sex, or treatment outcomes (cured, treatment completed, treatment failure, loss to follow-up, death, or not evaluated).

### Study design and sampling procedure

This multi-centered, facility based retrospective study was conducted in public health facilities of Arba Minch town, Southern Ethiopia. Between September 2021 and August 2024, a total of 700,340 patients were seen across these public health institutions for various health services. Of these, 675 patients were diagnosed with all forms of tuberculosis (TB) and initiated on anti-TB treatment at the three facilities (Fig 1). Among them, 609 patients fulfilled the eligibility criteria and were included in the study (Fig 1). To maximize representativeness, eliminate sampling error, and enhance the reproducibility of the findings, the study employed a census of all eligible TB patients during the study period.

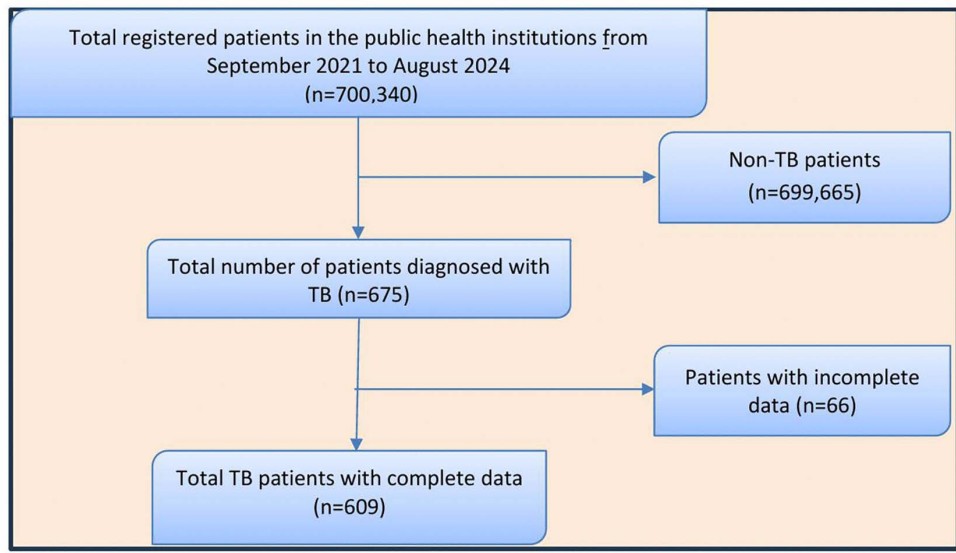

**Fig 1. Sampling procedure for the selection of tuberculosis patients with complete data.**

## Data collection and diagnosis

Data on treatment outcomes and independent variables were collected from TB clinic-based unit TB registers and patient medical charts. Data for this retrospective study were accessed from unit TB registers and medical charts between January 20 and February 20, 2025, for research purposes. The data collected included demographic information such as gender, age, weight, and body mass index (BMI). The study also collected many clinical characteristics such as the type of TB, TB registration category, GeneXpert *Mycobacterium tuberculosis*/rifampicin resistance (MTB/RIF) assay diagnosis results, HIV status, initiation of antiretroviral therapy (ART), and indications of presumptive drug resistance (PDR). During data collection experienced staff working in the TB treatment units were involved. They were trained and collected under the strict supervision of the principal investigator. The data were collected using a pretested structured data extraction questionnaire that included the above-mentioned information.

In these institutions, TB-suspected patients were first identified by examining the signs and symptoms, prior history of TB, and chest x-ray results. In addition, sputum samples were sent for a molecular drug susceptibility test to confirm the diagnosis and make sure they don't have a drug resistance pattern. The final diagnosis of TB was made according to the recommendations of the Ministry of Health (MoH) of Ethiopia [6]. Treatment was given based on national guidelines for TB treatment. This included a full course of anti-TB therapy for a total of six months. Two-month intensive phase with four major drugs (isoniazid, rifampicin, pyrazinamide, and ethambutol), and an additional four-month continuation phase with rifampicin and isoniazid. Outcomes were labeled successful (cure/complete) and unsuccessful (death, lost-to-follow-up, failure, not evaluated) [6].

## Operational definition

**Cured:** A TB patient who started treatment with bacteriologically verified TB and finished the treatment as advised by national policy with evidence of a bacteriological response but no signs of failure [16].

**Treatment completed:** A TB patient who followed the national policy suggested a course of therapy but whose results did not fulfill the criteria for cure or treatment failure [16].

**Treatment failure:** A TB patient whose treatment plan required being stopped or permanently switched to another treatment plan [16].

**Died:** A TB patient who passed away for whatever reason, either before beginning treatment or while receiving it [16].

**Lost to follow-up:** A TB patient who has not started therapy or whose regimen has been stopped for eight or more weeks in a row after starting treatment at least four weeks previously [16].

**Not evaluated:** A TB patient for whom no treatment result has been established. This applies to situations where the reporting unit is unsure of the treatment outcome and situations where the case has been transferred to another treatment facility [16].

**New patients:** Patients with TB who have never undergone TB therapy or who have just started receiving anti-TB drugs. New patients may have positive or negative bacteriology and may have disease in any area of the body [16].

**Previously treated:** A TB patient may have the disease at any anatomical site, positive or negative bacteriology, and have taken anti-TB medications for at least one month in the past [16].

**Successful treatment outcome:** If TB patients finished therapy with symptom clearance or were cured (i.e., had a negative smear microscopy at the end of treatment and on at least one prior follow-up test) [16].

**Unsuccessful treatment outcome:** If TB patients had treatment but had treatment failure (i.e., were still smear-positive after five months), were lost to follow-up (i.e., patients who stopped taking their medication for two or more months consecutively after registering), or died [17].

## Data quality control

To minimize potential biases, a pre-test was conducted with 5% of the final sample size before the main data collection, and the checklist was revised based on the findings to ensure completeness and clarity. The pre-tested, structured questionnaire, prepared in English, was checked for consistency after the pretest.

The data collectors and a supervisor were trained for two days on the purpose of the study and the data collection procedures. All finalized data were reviewed for completeness and clarity by the principal investigator before and during data management, storage, and analysis.

## Data processing and analysis

Data collected for this study were managed using EpiData version 3.1 software. The data were thoroughly checked for completeness, coded, and entered into the software. The data was then exported and analyzed using SPSS version 26.

Due to the dichotomous nature of the outcome variable (successful vs. unsuccessful treatment outcomes), logistic regression analysis was employed. This model was specifically chosen for its ability to estimate the odds of an event occurring while effectively adjusting for multiple covariates. Initially, a binary logistic regression analysis was performed to identify variables associated with the treatment outcome at a crude level. Any variables with a p-value of <0.25 in the binary analysis were considered a potential candidate for inclusion in the final (i.e., multivariable logistic regression) model to control for potential confounding and to identify independent predictors of treatment outcome. Statistical significance in the final model was established at a p-value of ≤0.05.

Finally, model fitness was assessed using the Hosmer–Lemeshow goodness-of-fit test, and multicollinearity among independent variables was checked to ensure the reliability of the final regression model.

## Ethical approval and consent to participants

The ethical approval to conduct this study was obtained from the Arba Minch University Ethical Review Board (IRB/23304/25), and a cooperation letter was given to the facilities. This study had no contact with patients, and the information obtained regarding patients was made anonymous.

## Result

### Socio-demographic characteristics of the participants

A total of 609 participants enrolled in this study were systematically analyzed from the total (700,340) population attending in the three health centers from September 2021 to August 2024. The majority of the participants were aged between 21 and 40 years, with 327 (53.7%) individuals falling within this age range. Additionally, 380 (62.4%) participants were male. More than half (53.9%) of the participants, were married, and over two-thirds (77.7%) participants had received formal education. Furthermore, 326 (54.6%) participants were classified as underweight (Table 1).

### Clinical and behavioral related characteristics

More than half (52.1%) of the participants were reported no history of alcohol consumption, while a greater (76.5%) proportion did not have a history of cigarette smoking. Most (71.8%) participants were diagnosed with pulmonary TB and greater than 90% were found new patients (Table 2). All participants were offered an HIV test; among them, the majority (78.2%) of the results were found non-reactive. Of those who tested reactive (21.8%), the vast majority (98.4%) were already began antiretroviral therapy. Additionally, 607 (99.6%) of the participants did not show signs of PDR, and 413 (67.8%) did not undergo the GeneXpert MTB/RIF test. 17 (2.8%) Patients who couldn't expectorate sputum, and debilitated HIV/AIDS patients with low cluster of differentiation (CD4) count were evaluated with urine Lateral Flow-Lipoarabinomanan test (LF-LAM). Furthermore, a significant number (n = 443) of participants had a positive smear test result (Table 2).

### TB treatment outcomes

According to this study, out of the 609 patients, 529(86.9%) TB patients had a successful treatment outcome. The cure rate for TB patients who received treatment was 39.1%, and the treatment completion rate was 47.15%. Whereas 58 patients (9.5%) died, 9 patients (1.5%) defaulted and the outcome of 13 patients (2.1%) was declared not evaluated making the total unsuccessful treatment outcome 13.1% (Fig 2).

### Factors associated with successful TB treatment outcome

Seven variables in binary logistic regression with a p-value of less than 0.25 became candidates for multiple logistic regressions (Table 3). In multiple logistic regressions, three variables were significantly associated (Table 3) with treatment

**Table 1. Socio-demographic characteristics of study participants.**

| Variables | Characteristics | Frequency | % |
|---|---|---|---|
| Age (year) | <20 | 129 | 21.2 |
| | 21-40 | 327 | 53.7 |
| | 41-60 | 111 | 18.2 |
| | >60 | 42 | 6.9 |
| Gender | Male | 380 | 62.4 |
| | Female | 229 | 37.6 |
| Marital status | Unmarried | 281 | 46.1 |
| | Married | 328 | 53.9 |
| Educational level | Informal education | 136 | 22.3 |
| | Formal education | 473 | 77.7 |
| Body mass index | Under weight | 326 | 54.6 |
| | Normal weight | 251 | 42.1 |
| | Overweight | 19 | 3.1 |

**Table 2. Clinical and behavioral characteristics of study participants.**

| Variables | Characteristics | Frequency | % |
|---|---|---|---|
| History of alcohol | Yes | 292 | 47.9 |
| | No | 317 | 52.1 |
| History of smoking | Yes | 143 | 23.5 |
| | No | 466 | 76.5 |
| Type of TB | Pulmonary TB | 437 | 71.8 |
| | Extra pulmonary TB | 156 | 25.6 |
| | Unknown | 16 | 2.6 |
| Category of TB | New | 566 | 92.9 |
| | Repeat | 37 | 6.1 |
| | Transfer in | 6 | 1.0 |
| HIV comorbidity | Reactive | 133 | 21.8 |
| | Non-reactive | 476 | 78.2 |
| ART initiation | Yes | 131 | 98.5 |
| | No | 2 | 1.5 |
| PDR | Yes | 2 | 0.4 |
| | No | 607 | 99.6 |
| GeneXpert MTB/RIF assay performed | Yes | 179 | 29.4 |
| | No | 413 | 67.8 |
| | LF-LAM | 17 | 2.8 |
| Smear examination | Positive | 443 | 72.7 |
| | Negative | 162 | 26.6 |
| | Not done | 4 | 0.7 |

TB = Tuberculosis, HIV = Human immunodeficiency Virus, ART = Antiretroviral therapy, PDR = Presumptive drug resistance, LF-LAM = Later flow Lipoarabinomannan.

outcome of TB with p-value <0.05. These are educational status (p = 0.028), being unmarried (p = 0.023), and having extra-pulmonary TB (p = 0.017) (Table 3).

Those study participant who had formal education were 2 times more likely to have successful treatment outcome (AOR = 1.67; 95% CI = 1.05–2.66; p = 0.028) than their counterparts. Those participants who were unmarried during the time of treatment were 2 times more likely to have successful treatment outcome (AOR = 1.64; 95% CI = 1.05–2.47; p = 0.023) than married participants and participants with EPTB were almost 4 times more likely to have successful treatment outcome (AOR = 3.8; 95% CI = 1.23–11.7; p = 0.017) than participants with pulmonary TB.

## Discussion

Globally, TB is a significant public health issue, with one of the highest morbidity and mortality rates. It is estimated that millions of new cases arise each year, particularly in regions with limited healthcare resources, where factors like poverty, malnutrition, and the prevalence of HIV/AIDS further exacerbate the situation [18,19]. In this study, treatment outcome and factors associated with TB were assessed. The magnitude of successful treatment outcomes among TB patients was found to be 86.9%, which is a bit lower than the world (88%) and African (88%) average TB treatment success rate for new and relapse cases reported in 2024 [1]. However, this result is nearly similar to studies conducted in Gibe Woreda, Southern Ethiopia (85.5%) [12] and Debre Tabor Town, Northwestern Ethiopia (87.1%) [20]. Such consistency in outcomes could suggest comparable healthcare practices or disease management strategies in these regions (Table 4).

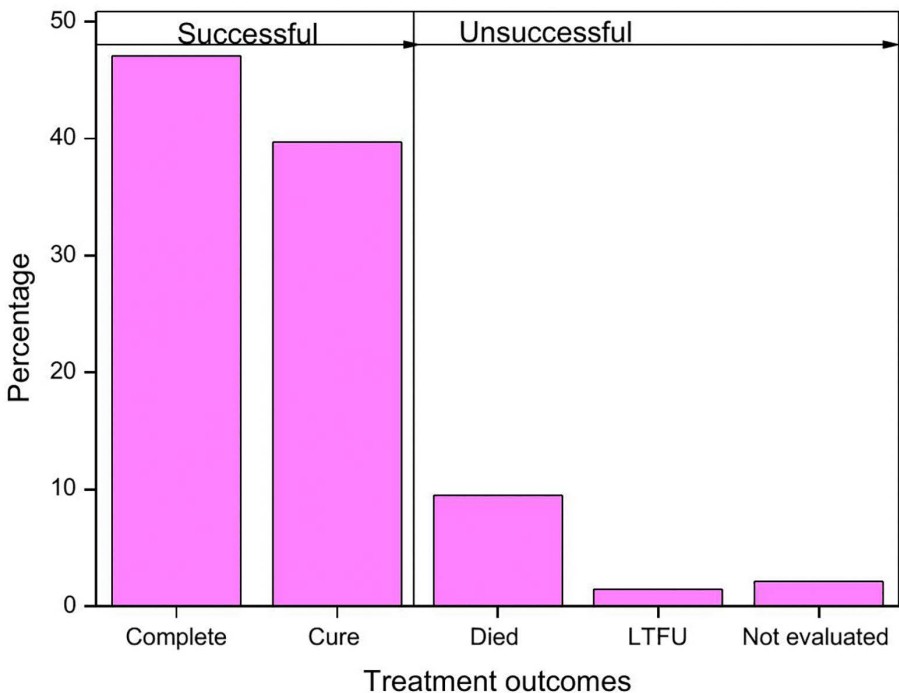

**Fig 2. Treatment outcome of TB patients treated at public health institutions of Arba Minch town.**

**Table 3. Factors associated with tuberculosis treatment outcome in public health institutions of Arba Minch town (n = 609).**

| Variables | Category | TB-TO | | COR (95% CI)ᵃ | P-value | AOR (95% CI)ᵇ |
|---|---|---|---|---|---|---|
| | | Successful | Unsuccessful | | | |
| Educational Status | Informal education | 100 | 36 | 1 | | 1 |
| | Formal education | 380 | 93 | 0.68(0.43-1.05) | 0.028 | 1.67(1.05-2.66) *** |
| Marital status | Unmarried | 210 | 71 | 1.57(1.06- 2.32) | 0.023 | 1.64(1.05-2.47) *** |
| | Married | 270 | 58 | 1 | | 1 |
| Alcohol history | Yes | 237 | 55 | 1.31(0.88-1.94) | 0.174 | 1.08(0.70-1.67) |
| | No | 243 | 74 | 1 | | 1 |
| Smoking history | Yes | 361 | 105 | 1 | | 1 |
| | No | 119 | 24 | 0.69(0.42-1.13) | 0.143 | 1.40(0.81-2.42) |
| Type of TB | Pulmonary | 335 | 102 | 1 | | 1 |
| | Extra pulmonary | 135 | 21 | 3.85(1.26-11.77) | 0.017 | 3.80(1.23-11.7) *** |
| | Unknown | 10 | 6 | 1.97(0.69-5.55) | 0.199 | 2.00(0.69-5.74) |
| HIV-Coin | Yes | 113 | 20 | 1.67(0.99-2.82) | 0.052 | 0.62(0.36-1.06) |
| | No | 367 | 109 | 1 | | 1 |
| ART-Init | Yes | 111 | 20 | 1.63(0.97-2.76) | | 1.89(0.14-24.5) |
| | No | 369 | 109 | 1 | | 1 |

TB-TO=TB treatment outcome, COR = Crude Odd Ratio, AOR = Adjusted Odds Ratio, HIV-Coin = Human Immunodeficiency Virus Co-infection, ART-Init = Antiretroviral initiation, *** = p < 0.05. Values in a column indicated in letters "a" and "b" show the bivariate and multivariate logistic regression results respectively.

**Table 4. Comparing TB treatment outcome and associated factors in prior studies.**

| Study area | Sample size (number) | Methodology | TSR (%) | Associated factors | Reference |
|---|---|---|---|---|---|
| Arba Minch, Ethiopia | 609 | Retrospective | 86.9 | Formal education, unmarried, and EPTB (+) | This study |
| Gondar, Ethiopia | 1584 | Retrospective cross-sectional | 60.1 | Age 30–39.9, smear negative, and being EPTB, retreatment, HIV positive (-) | [21] |
| Woldia, Ethiopia | 270 | Cross-sectional | 80.7 | Older age, female sex, year of registration in 2015, and being HIV positive (-) | [22] |
| Afar, Ethiopia | 380 | Retrospective | 81.8 | Older age, HIV Reactive status (-); while (four-week attendance, sputum follow-up test, and being female (+) | [23] |
| Wolaita, Ethiopia | 232 | Retro-cohort | 82.5 | No statistically significant associated factor | [10] |
| Gibe, Ethiopia | 400 | Cross-sectional | 85.5 | Age greater than 45, male sex, travel distance over 10 km, lack of family support, bedridden functional status (-) | [12] |
| Debretabor, Ethiopia | 303 | Retrospective | 87.1 | Female sex, rural residence, smear negative at 2nd month of treatment (-) | [20] |
| Jimma, Ethiopia | 1249 | Retrospective | 88.2 | Smear-negative, Extra-pulmonary TB, unknown HIV status (-) | [24] |
| Harare, Ethiopia | 1236 | Retrospective | 92.5 | Female, HIV negative, weight 20–29 kg (+); while Age > 54 (-) | [8] |
| Enfranz, Ethiopia | 417 | Retrospective | 94.8 | No statistically significant associated factor | [25] |
| Easter Cape, South Africa | 427 | Ambidirectional | 65.8 | HIV co-infection, history of smoking (-) | [26] |
| Ashanti region, Ghana | 891 | Retrospective | 68.4 | Retreatment (-); while Age less than 20 or age range b/n 51–60, pre-treatment weight > 35 kg (+) | [20] |
| Anambra & Oyo, Nigeria | 1281 | Cross-sectional | 75.8 | Geographical location, facility tier/type (-) | [27] |
| Mogadishu, Somalia | 385 | Cross-sectional | 81.5 | Married, educated, HIV-negative, new treatment, knowledge of TB (+) | [28] |
| Hunan province, China | 308,860 | Retro-cohort | 98.6 | Age, male, being severally ill, TB treatment history, year of diagnosis (-) | [29] |
| Ahmedabad, India | 8301 | Retro-cohort | 87.9 | Age 45–54 (-) | [30] |
| Ethiopia* | NS | Survey report | 86.0 | Age, gender, HIV, under-nutrition, diabetes mellitus, alcohol, drug resistance, smoking, poverty (-) | [1] |
| Africa* | NS | Survey report | 88.0 | Age, gender, HIV, under-nutrition, diabetes mellitus, alcohol, drug resistance, smoking, poverty (-) | [1] |
| World* | NS | Survey report | 88.0 | Age, gender, HIV, under-nutrition, diabetes mellitus, alcohol, drug resistance, smoking, poverty (-) | [1] |

(+)= Significantly associated with successful treatment outcomes, (-)=Significantly associated with unfavorable treatment outcomes, EPTB = Extra-pulmonary tuberculosis, TSR: = Treatment success rate, Retro-cohort = Retrospective cohort, *=Average report in 2024, NS = Not stated, Prosp-cohort = Prospective cohort.

In contrast to our finding, studies from various locations have reported significantly lower TB treatment success rates in Ethiopia and other countries in Africa. For example, a study in Ghana (five directly observed therapy short-course (DOTS) centres at the Atwima Nwabiagya district) found a lower TB treatment success rate (68.5%) [31], while in South Africa (Eastern Cape hospitals) recorded an even lower (65.8%) rate of treatment success [26]. Similarly, studies in two Nigeria states (Anambra and Oyo) and Somalia (across seven public TB management units in Mogadishu) demonstrated a lower TB treatment success rate of 75.8% [27] and 81.8% [28], respectively. Lower TB treatment success rates were also observed across various regions in Ethiopia, highlighting significant internal differences that may reflect localized challenges in healthcare delivery. Specifically, TB treatment success rates reported in the Afar region (81.8%) [32], Gonder (60.1%) [21], Wolayta (82.5%) [10], and Woldia (80.7%) [22] (Table 4). These regional variations may be attributed to several factors, including differences in study periods and sample sizes. The discrepancies also highlight how regional

healthcare systems and patient demographics ultimately influence adherence and treatment completion. A comprehensive understanding of these discrepancies is essential for designing effective TB control strategies across the country.

On the other hand, the current study TB treatment success rate was lower than those reported in Harar, Eastern Ethiopia (92.5%) [8], Jimma, Southwestern Ethiopia (88.3%) [24], and Enfraz Health Center, Northwestern Ethiopia (94.8%) [25] (Table 4). The higher success rates in these studies may be attributed to different methodologies, particularly the use of larger sample sizes, which typically enhance the reliability of findings. Moreover, the types of healthcare facilities and their level may significantly influence treatment outcomes. Factors such as the demographic characteristics of the patient population and the degree of urbanization in the settings could also play a crucial role in affecting these outcomes. In addition, our treatment success rate is lower than that reported in Hunan Province, China (98.6%) [29] as well as the national average of 95% in 2022 [1]. This difference may be attributed to China's strong universal health coverage, substantial TB financing, and the consistent application of DOTS through its National TB Control Program. Similarly, in India 87.9% of TB treatment success rate were reported in 2023 [30], supported by its Revised National TB Control Program, which emphasizes early case detection, patient-centered treatment, and enhanced monitoring systems.

Multivariate logistic regression model analysis showed that participants who were unmarried during the time of treatment had 64% higher odds of having a successful treatment outcome (AOR = 1.64; 95% CI = 1.05–2.47; p = 0.017) compared to those who were married. This finding contrasts with previous studies, which demonstrated unmarried individuals had a lower social support system [10,22,28]. The most likely explanation could be due to married individuals in the Gamo zone may face competing family and household responsibilities, such as caregiving, income-generating activities, and maintaining household obligations, which could interfere with adherence and clinic attendance. Indeed, other studies in Southern Ethiopia found that being married was associated with lower odds of successful TB treatment outcome [33]. A study conducted within Gamo Gofa zone (a similar study area for our study) also highlighted that family dynamics, such as a patient's failure to disclose their TB status to household members, were strongly associated with non-adherence [34]. Based on our finding and these evidences [33,34], the burden of marital and household commitments are critical contributing factors to suboptimal treatment success in this specific context, suggesting the vital need for future research to deeply explore and mitigate these underlying mechanisms. Our study is deficient in addressing how factors such as socioeconomic status, cultural norms and stigma, and household dynamics affect the treatment outcome among married patients. Future studies should explore a longitudinal or mixed-methods design to explore how the interplay of those factors and marital status affect the treatment of TB.

The educational status of the participants showed statistically significant association with treatment outcomes, particularly those who attended formal education were almost twice as likely to achieve a successful treatment outcome (AOR = 1.67; 95% CI = 1.05–2.66; p = 0.028) compared to those without formal education. This finding is consistent with a study conducted in Somalia [28]. Various pieces of evidence indicate that individuals with higher levels of education tend to have better health literacy, medication adherence, improved health-seeking behavior, and the overall socioeconomic advantages that education provides [35–37].

Participants with EPTB were almost 4 times more likely to have successful treatment outcome (AOR = 3.8; 95% CI = 1.23–11.7, p = 0.023) than participants with pulmonary TB. This finding contrasts with most of the previous studies [8,10,21,22,24,28]. The possible explanation might be some of the EPTB infections such as infections involving lymph nodes may have a more localized and less aggressive presentation compared to pulmonary TB, EPTB patients might have stronger immune reactivity when the infection is confined to specific sites, and patients with pulmonary TB might face additional challenges due to respiratory complications such as respiratory failure, which can complicate treatment and lead to poorer outcomes. Previous studies demonstrated variable results on the treatment success rate of EPTB, which is dependent on different factors such as location of the infection, HIV co-morbidity, and age [38,39]. Those patients presenting with EPTB, particularly involving the pleura, lymph nodes, and abdomen, had better outcomes. However, those

infections involving the central nervous system carry poor outcomes [38]. Since our data couldn't identify the specific types of EPTB, future studies needed that focus on incorporating data on specific types of EPTB in their studies.

Although many studies [1,8,22] stated that HIV positive, advanced age, male sex, and malnutrition are commonly associated with unfavorable TB treatment outcomes, our study did not find significant associations with these factors. The finding that HIV positive status was not statistically associated with poor TB treatment outcomes in our study, contrary to global trends, likely originates from the effectiveness of the local integrated TB-HIV collaborative services. The most probable primary reason is the high (98.5%) ART coverage and timely initiation (Table 2) among TB-HIV co-infected patients. Antiretroviral therapy rapidly suppresses the HIV virus, leading to immune reconstitution, which allows the patient's body to respond effectively to the anti-TB drugs, mitigating the risk of death and treatment failure typically seen in immunocompromised individuals [1].

The prominent consumption of low-fat staple foods in the study area like *Enset* (including its products: *Kocho*, *Bulla*, and *Godere*), may directly contribute to lower population BMI values. Since BMI is a simple measure that does not differentiate between lean muscle mass and fat stores, a low BMI in this context may primarily reflect limited dietary fat intake rather than severe protein-energy malnutrition. Consequently, BMI may be an inadequate indicator of specific nutritional deficits (e.g., in essential fatty acids or fat-soluble vitamins) crucial for immune function and recovery. This limitation likely accounts for the observed lack of association between BMI and the measured TB treatment outcomes in this population.

To figure out the current study TB treatment outcome from other previous TB studies, a comparative Table (Table 4) is prepared.

Our study has limitations. It is based on retrospective analysis of medical records, which were not designed for research, which lacked data on factors such as level of income, social and cultural roles of married individuals, accessibility to health care, specific types of EPTB and longitudinal insights regarding the challenges faced by TB patients. Despite these limitations, our study provides information about the treatment outcome and associated factors among TB patients in Arba Minch town where data has not been previously published.

## Conclusion

In this study, the overall TB treatment success rate was found to be 86.9%, which is significantly associations with being unmarried, attending formal education, and being diagnosed with EPTB. This finding is below the target set by the WHO end TB strategy, which aims to achieve a TSR of 90% by 2030. This indicates that attention should be given to patients who are married, individuals with low educational levels, and patients diagnosed with pulmonary TB in order to improve the overall outcome of TB treatment in the Gamo zone. While the study benefits from being multi-centered and large sample size, future studies should focus on longitudinal studies to identify other potential socioeconomic and cultural factors that could affect the treatment outcomes of TB patients.

## Acknowledgments

The authors are grateful for the data collectors, hospital staff working at the TB clinics, and heads of hospitals for their willingness and unreserved contribution for making this study fruitful.

## Author contributions

**Conceptualization:** Behailu Asmamaw, Rediet Yifru Mamo, Yohannes Chemere Wondmeneh, Dagmawi Nega Shibeshi, Eunice Borkor Bortequaye, Temesgen Esayas Bolliso, Tiwabwork Tekalign.

**Data curation:** Behailu Asmamaw, Rediet Yifru Mamo, Berhanu Amare Taye, Yohannes Chemere Wondmeneh, Tiwabwork Tekalign.

**Formal analysis:** Behailu Asmamaw, Temesgen Teklu Ziza.

**Investigation:** Abrham Workineh Azale, Dagmawi Nega Shibeshi.

**Methodology:** Behailu Asmamaw, Temesgen Tamiru Tadese, Rediet Yifru Mamo, Berhanu Amare Taye, Abrham Workineh Azale, Yohannes Chemere Wondmeneh, Eunice Borkor Bortequaye, Temesgen Teklu Ziza, Temesgen Esayas Bolliso.

**Software:** Behailu Asmamaw.

**Writing – original draft:** Behailu Asmamaw, Temesgen Tamiru Tadese, Berhanu Amare Taye, Abrham Workineh Azale, Dagmawi Nega Shibeshi, Eunice Borkor Bortequaye, Temesgen Teklu Ziza, Temesgen Esayas Bolliso, Tiwabwork Tekalign.

**Writing – review & editing:** Behailu Asmamaw, Eunice Borkor Bortequaye, Tiwabwork Tekalign, Awoke Guadie.

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
