## [Decision Letter · Decision Letter 0]

18 Aug 2025

Dear Dr. Guadie,

Thank you for submitting your manuscript to PLOS ONE. After careful consideration, we feel that it has merit but does not fully meet PLOS ONE’s publication criteria as it currently stands. Therefore, we invite you to submit a revised version of the manuscript that addresses the points raised during the review process.

Please submit your revised manuscript by Oct 02 2025 11:59PM. If you will need significantly more time to complete your revisions, please reply to this message or contact the journal office at plosone@plos.org . A rebuttal letter that responds to each point raised by the academic editor and reviewer(s). You should upload this letter as a separate file labeled 'Response to Reviewers'.A marked-up copy of your manuscript that highlights changes made to the original version. You should upload this as a separate file labeled 'Revised Manuscript with Track Changes'.An unmarked version of your revised paper without tracked changes. You should upload this as a separate file labeled 'Manuscript'.

We look forward to receiving your revised manuscript.

Kind regards,

Frederick Quinn

Academic Editor

PLOS ONE

2. In the online submission form, you indicated that [The data underlying the results presented in the study are available from the corresponding author with a reasonable request.].

Additional Editor Comments (if provided):

Reviewers' comments:

Reviewer's Responses to Questions

**Comments to the Author**

1. Is the manuscript technically sound, and do the data support the conclusions?

Reviewer #1: Yes

Reviewer #2: Yes

2. Has the statistical analysis been performed appropriately and rigorously?

Reviewer #1: No

Reviewer #2: Yes

3. Have the authors made all data underlying the findings in their manuscript fully available?

Reviewer #1: Yes

Reviewer #2: Yes

4. Is the manuscript presented in an intelligible fashion and written in standard English?

Reviewer #1: Yes

Reviewer #2: Yes

Reviewer #1: The authors should work on describing the methodological procedures in more detail for the sake of reproducibility of this study. By clearly mentioning the inclusion criteria, sampling methods, how missing data were handled, and how potential biases were addressed.

Please provide justifications for analytical approaches. Please provide tools for analysis of predictors of mortality.

The findings listed in this research article mainly reiterate the known risk and known associations, and lacks in providing actionable new insights.

Discussion should be improved by addressing the global scenarios beyond Africa, including other countries like China, India.

Addressing why HIV positivity, BMI are not a significant associations.

Reviewer #2: Summary:

The manuscript investigated the outcome of tuberculosis treatment and associated factors in multicenter facilities. The research being done in multicenter make the results more representative of broader population. The study is clearly presented and the methodology meets the PlOS One guidelines. The manuscript is technically sound, and the data supports the conclusions.

Overall impression: the study addresses an interesting research question. Having said these, few minor issues must be addressed before publication.

1. In the title of the study:

• “cross-sectional study” must be included in the title

2. In the abstract:

• In the abstract part (line 31 to 33) and the on results (line 226 and 227); based on your results and table 3, educational status was not statistically significant. And the variables which are associated with successful treatment outcome should be written as “being unmarried” and “extra-pulmonary TB” instead of marital status and type of TB.

3. In the introduction part:

• Write the reference of line 52 to 54.

4. In the methodology part:

• Language and grammar, line 128 to 130 is not written as a sentence, please rewrite it.

5. In results and discussion part:

I. Line 224, the subtitle should be written as “factors associated with successful tuberculosis treatment outcome” since the study is assessing the associated factors of successful treatment outcome instead of factors associated with tuberculosis treatment outcome.

II. Line 226 and 227; education is not significant from your table (table 3). The texts “marital status” and “type of TB” should be written as being unmarried, and having extra-pulmonary TB as they are the ones which was associated with successful treatment outcome. In addition, use past tense when you talk about your results.

III. Being never married will be difficult to find from medical records it would be better if it is written as unmarried during the time of treatment.

IV. On discussion part; when you discuss researches done in Ethiopia please mention Ethiopia in addition to the name of the place so that the international readers will know it is Ethiopia.

V. On discussion part on line 271 to 274 you wrote “married individuals in Arba Minch area share

multiple competing sociocultural obligations, such as workloads to maintain family incomes and family responsibilities that could detract them from their health management efforts, unlike never-married individuals who may prioritize their health more effectively”. How do you reach to that conclusion?

**Do you want your identity to be public for this peer review?** For information about this choice, including consent withdrawal, please see our Privacy Policy

Reviewer #1: No

Reviewer #2: **Yes: ** Liwam Kidane Gezahegn

---

## [Author Response · Author response to Decision Letter 1]

30 Oct 2025

All comments from the editor and reviewers have been carefully addressed. Revised files have been uploaded as requested. We appreciate your consideration and look forward to the next steps in the review process.

---

## [Decision Letter · Decision Letter 1]

23 Nov 2025

Magnitude of tuberculosis treatment outcomes and associated factors in public health institutions of Arba Minch town, Southern Ethiopia: A multi-centered retrospective cross-sectional study

PONE-D-25-35010R1

Dear Dr. Guadie,

We’re pleased to inform you that your manuscript has been judged scientifically suitable for publication and will be formally accepted for publication once it meets all outstanding technical requirements.

Kind regards,

Frederick Quinn

Academic Editor

PLOS ONE

Additional Editor Comments (optional):

Reviewers' comments:

Reviewer's Responses to Questions

**Comments to the Author**

Reviewer #1: All comments have been addressed

Reviewer #2: All comments have been addressed

2. Is the manuscript technically sound, and do the data support the conclusions?

Reviewer #1: Yes

Reviewer #2: Yes

3. Has the statistical analysis been performed appropriately and rigorously?

Reviewer #1: Yes

Reviewer #2: Yes

4. Have the authors made all data underlying the findings in their manuscript fully available?

Reviewer #1: Yes

Reviewer #2: (No Response)

5. Is the manuscript presented in an intelligible fashion and written in standard English?

Reviewer #1: Yes

Reviewer #2: Yes

Reviewer #1: Expand WHO in introduction line 46

Add reference for line 45-46

remove tuberculosis in line 120 (Study design section)

In clinical and behavioral related characteristics please adopt a uniform format of answering: mention Smear Test positive individuals in (); Same in TB treatment outcomes unsuccessful treatments

it would be better to expand on HIV and TB relations (up to authors)

Reviewer #2: (No Response)

**Do you want your identity to be public for this peer review?** For information about this choice, including consent withdrawal, please see our Privacy Policy

Reviewer #1: No

Reviewer #2: **Yes: ** Liwam Kidane Gezahegn

---

## [Editor Report · Acceptance letter]

PONE-D-25-35010R1

PLOS ONE

Dear Dr. Guadie,

I'm pleased to inform you that your manuscript has been deemed suitable for publication in PLOS ONE. Congratulations! Your manuscript is now being handed over to our production team.

Kind regards,

on behalf of

Dr. Frederick Quinn

Academic Editor

PLOS ONE